# Antidepressant Medication Does Not Contribute to the Elevated Circulating Concentrations of Acylethanolamides Found in Substance Use Disorder Patients

**DOI:** 10.3390/ijms241914788

**Published:** 2023-09-30

**Authors:** Jesús Herrera-Imbroda, María Flores-López, Nerea Requena-Ocaña, Pedro Araos, Nuria García-Marchena, Jessica Ropero, Antonio Bordallo, Juan Suarez, Francisco J. Pavón-Morón, Antonia Serrano, Fermín Mayoral, Fernando Rodríguez de Fonseca

**Affiliations:** 1Instituto de Investigación Biomédica de Málaga y Plataforma en Nanomedicina—IBIMA Plataforma Bionand, 29590 Málaga, Spain; jesus.herrera.imbroda.sspa@juntadeandalucia.es (J.H.-I.); maria.flores@ibima.eu (M.F.-L.); nerea.requena@ibima.eu (N.R.-O.); paraos@uma.es (P.A.); ngmarchena@ucm.es (N.G.-M.); jessica.ropero@ibima.eu (J.R.); juan.suarez@uma.es (J.S.); javier.pavon@ibima.eu (F.J.P.-M.); fermin.mayoral.sspa@juntadeandalucia.es (F.M.); 2Unidad de Gestión Clínica de Salud Mental, Hospital Regional Universitario de Málaga, 29010 Málaga, Spain; antonio.bordallo.sspa@juntadeandalucia.es; 3Departamento de Farmacología y Pediatría, Facultad de Medicina, Universidad de Málaga, 29071 Málaga, Spain; 4Departamento de Psicología Básica, Facultad de Psicología, Universidad de Málaga, 29071 Málaga, Spain; 5Departamento de Psicobiología y Metodología, Facultad de Psicología, Universidad Complutense de Madrid, 28223 Madrid, Spain; 6Departamento of Anatomía, Medicina Legal e Historia de la Ciencia, Facultad de Medicina, Universidad de Málaga, 29071 Málaga, Spain; 7Unidad Clínica Área del Corazón, Hospital Universitario Virgen de la Victoria, 29010 Málaga, Spain; 8Centro de Investigación Biomédica en Red de Enfermedades Cardiovasculares (CIBERCV), Instituto de Salud Carlos III, 28029 Madrid, Spain; 9Unidad Clínica de Neurología, Hospital Regional Universitario de Málaga, 29010 Malaga, Spain; 10Andalusian Network for Clinical and Translational Research in Neurology (NEURO-RECA), 29001 Malaga, Spain

**Keywords:** substance use disorders, biomarkers, acylethanolamides, antidepressants, psychiatric co-morbidity, depression

## Abstract

Circulating acylethanolamides (NAEs) are bioactive signaling molecules that modulate multiple homeostatic functions including mood and hedonic responses. Variations in their plasma concentrations are associated with substance use disorders (SUD) and recent studies suggest that psychotropic medication might influence its circulating levels, limiting its use as a clinical biomarker of addiction. In addition, they might have a role as mediators of the pharmacological effects of psychotropic drugs. Thus, in mild depression, the response to selective serotonin reuptake inhibitor-type antidepressants (SSRI) is associated with a marked increase in circulating NAEs. To further investigate if antidepressants are able to modify the plasma concentration of NAEs in SUD patients, we analyzed the circulating levels of NAEs in 333 abstinent and 175 healthy controls on the basis of the treatment with SSRI antidepressants. As described previously, SUD patients display higher concentrations of NAEs than those measured in a control population. This increase was not further modified by antidepressant therapy. Only marginal increases in palmitoylethanolamide (PEA), oleoylethanolamide (OEA), or docosatetraenoyl-ethanolamide (DEA) were found, and the net effect was very small. Thus, our study shows that treatment with SSRI-type antidepressants does not modify the clinical utility of monitoring enhanced NAE production as biomarkers of SUD. In addition, the possibility that a blunted NAE response to antidepressant therapy might be related to the loss of efficacy of SSRIs in dual depression emerges as an attractive hypothesis that needs to be addressed in future studies.

## 1. Introduction

Substance use disorder (SUD) is a chronic relapsing mental disorder characterized by compulsive drug seeking despite the negative consequences associated with excessive drug intake [1,2]. SUD is often associated with multiple medical and psychiatric comorbidities that complicate diagnosis and clinical outcomes [3]. The pathophysiology of SUD is complex and involves various brain anatomical structures and several neurobiological circuits [4]. One of the biological systems involved in the etiopathogenesis and natural history of drug addiction is the endocannabinoid system (EC) [5,6]. It is defined as a neuromodulatory system composed of bioactive lipids (endocannabinoids and related *N*-acylethanolamines (NAEs)) involved in the control of multiple homeostatic functions that include neural development, reward regulation, pain perception, or behavioral responses. This is achieved through endocannabinoid-derived molecular control of bioenergetics, synaptic transmission, and plasticity/inflammation/repair processes [7,8]. Acylethanolamides or *N*-acylethanolamines (NAEs) are bioactive lipids that include the true endocannabinoid anandamide (AEA) and other anandamide-like compounds capable of interfering with the actions of the true endocannabinoids since they share with AEA the same synthesizing and degrading enzymatic machinery. Additionally, NAEs retain the ability to modulate cannabinoid receptor signaling [9,10]. These bioactive lipids include oleoylethanolamide (OEA), palmitoylethanolamide (PEA), linoleoylethanolamide (LEA), and stearoylethanolamide (SEA) [7,11]. They can exert various biological, homeostatic, and anti-inflammatory functions through interaction with receptors such as the PPARα nuclear receptor [12], the ionotropic Vanilloid VR1 receptor [13] or the orphan receptors GPR119 and GPR55 [14,15]. The homeostatic roles of NAEs and the ability to modulate reinforcing processes [16] have set in place their potential role as biomarkers in chronic disorders, including addiction [17]. A constant replicated finding is an increase in the circulating concentrations of acylethanolamides in SUD patients, with the exception of cannabis consumption [18]. This elevation in circulating NAEs has been interpreted as a general adaptive response to the homeostatic dysbalance imposed by the pharmacological actions of abused drugs and the associated inflammatory response [6,17,18].

However, there are multiple factors that can modulate the NAEs’ response to abused drugs, especially psychotropic medication used for the treatment of SUD-associated psychiatric comorbidity. In fact, the co-existence of drug addiction and an additional psychiatric diagnosis is gaining presence in patients demanding treatment for uncontrolled drug use. This special comorbidity has come to be called “dual diagnosis”, a reality that has a high prevalence, a significant degree of clinical and social severity, and that usually associates a worsening of the prognosis and an increase in associated health costs [19,20,21]. Furthermore, in patients with these concurrent psychiatric and substance problems, a challenge for effective and rational treatment planning has been described [22]. Thus, while approved treatments for SUD are often underused [23], it is not uncommon for these patients to use psychotropic drugs approved for other indications, such as antidepressants or antipsychotics, due to their effects on comorbid symptoms or their ability to reduce dropouts [24,25]. Curiously both NAE production and EC activity can be triggered by the modulation of monoaminergic transmission (dopamine/serotonin) which is the main target for many of the psychotropic medications used in psychiatry [26,27,28]. Thus, the clinical relevance of using plasma NAE concentrations as biomarkers of SUD or dual diagnosis might be potentially limited by the interference generated by psychiatric medication. We have tested this hypothesis in a previously published work, where we have reported that, in patients with SUD, prescription of antipsychotic medication is associated with a greater increase in plasma concentrations of NAEs than that already found in patients with substance use disorder who do not take antipsychotic medication [29]. This is a relevant finding because of the generally accepted role of NAEs as physiological mediators of repair. The prescription of neuroleptics as antipsychotic medication might result in enhanced release of NAEs, eventually contributing to improving disease-associated symptoms and eventually helping to achieve healing, if therapeutic success is eventually achieved. This finding could also support a recent neurobiological model that we have proposed for psychosis and drug abuse, suggesting a common origin for both through dysregulation of pro/anti-inflammatory pathways where Ecs (and their regulation by antipsychotics) plays an important role [30].

In the case of antidepressants, a priori, we can expect a similar response, since an increase in circulating NAEs, especially OEA, has been described in depressed patients treated with classical selective serotonin reuptake inhibitors (SSRI) [31]. The potential contribution of OEA to the antidepressant effect of SSRI is supported by the well-known antidepressant action of OEA thanks to its modulatory effect of neuroinflammation [32,33]. However, we do not have a confirmation of whether SSRIs are able to modulate the circulating concentrations of NAEs in dual depression, the most frequent SUD-associated psychiatric co-morbidity. And this is relevant because in substance use disorder, SSRIs, especially if alcohol is one of the abused drugs, are much less efficient [19], despite being frequently prescribed [34,35]. Additionally, both endocannabinoids/NAEs and antidepressants converge on the dysregulated biological mechanisms underlying addiction [6,36].

Therefore, in order to better understand the role of NAEs in dual depression, as well as to further delimitate the potential clinical utility of monitoring circulating concentrations of NAEs in complicated SUD, we designed the present cross-sectional study. Its aim is to analyze the effect of antidepressant treatment on plasma concentration of NAEs in a cohort of SUD outpatients (mainly patients demanding treatment for alcohol or cocaine use disorders), differentiating whether or not they take prescribed antidepressant medication (SSRI), and comparing them with a control population.

## 2. Results

### 2.1. Sociodemographic Characteristics and Plasma Concentration of Acylethanolamides of Control and SUD Populations

A sociodemographic description of the total sample (N = 508) and differences found in plasma concentration of acylethanolamides of control and SUD populations can be consulted in a previously published work [29], as well as in the Appendix A.

### 2.2. Characteristics of the SUD Group Based in Antidepressant Treatment: Impact on Plasma Concentrations of Acylethanolamides

Table 1 shows a sociodemographic and clinical description of the 333 participants with SUD included in this study, differentiated based on the use of antidepressant treatment (SUD and SUD + SSRI antidepressant groups). An additional description of the total sample of SUD patients based on the primary drug they consulted for (alcohol or cocaine) is shown in Appendix A.

Patients in the SUD group had a mean age of 43.81 years with a standard deviation of 11.22, a mean BMI of 26.15 kg/m^2^ with a standard deviation of 4.79, and 84.1% of them were men. This overrepresentation of men with respect to women is a common finding in cross-sectional studies of the SUD population, reflecting profound gender differences in both, SUD prevalence and treatment demand. In the SUD + SSRI antidepressant group, the mean age of patients was 43.03 years with a standard deviation of 11.10, a mean BMI of 26.67 kg/m^2^, a standard deviation of 4.96, and 73.5% were men. Psychiatric comorbidity was more prevalent in the SUD + SSRI antidepressant group for all categories: mood disorders, anxiety disorders, psychotic disorders, personality disorders, ADHD, and two or more psychiatric disorders. A higher prevalence of SUD was registered in the group of SUD + SSRI antidepressants for alcohol, cannabis, and two or more substances. The use of psychotropic medication was more frequent in the SUD + SSRI antidepressant group for antipsychotics, anxiolytics, and disulfiram. Clear gender differences we observed between SUD and SUD + SSRI antidepressant treatment with respect to sex psychiatric comorbidity. These gender differences were more significant for mood disorders and >2 psychiatric disorders (*p* < 0.001). Polyconsumption differences were significant for cannabis use disorders (*p* ≤ 0.05). Overall, patients with SUD and antidepressant treatment have more psychiatric disorders than those without antidepressant therapy. Women showed more prevalence in antidepressant treatment than men, being prescribed more psychiatric medication.

Regarding the circulating concentrations of NAEs, Appendix A depicts the general increase in NAEs found in SUD patients, and previously reported. When the SUD population was stratified on the basis of treatment with antidepressants, initial analysis (Table 2 and Appendix A) shows that the concentration of NAEs was similar in both groups, with the exception of DEA and PEA, which were slightly higher in the SUD + SSRI antidepressants group. This suggests that antidepressant treatment did not generate the marked increase in circulating levels of NAEs previously described in depressed patients treated with SSRI attending primary care settings [31].

Since sex, age, and BMI might exert an influence on plasma concentrations of NAEs, we further analyzed the concentration of acylethanolamides between groups with one-way ANCOVA. Raw data for plasma concentrations of acylethanolamides were log10-transformed to ensure statistical assumptions of the one-way ANCOVA while controlling for age, BMI, and sex. Again, the analysis does not reveal a significant main effect of the group factor (SUD or SUD + SSRI antidepressant treatment) on any species of NAE concentrations (Appendix A), although marginally significant differences were found in the case of DEA [F (1.250) = 3.842, *p* = 0.051] and OEA [F (1.306) = 3.43, *p* = 0.065] (Figure 1).

### 2.3. Plasma Concentrations of Acylethanolamides Based on Sex and Comorbid Use of Cannabis

Table 3 and Table 4 show the concentrations of NAEs in the two comparison groups (SUD and SUD + SSRI antidepressant) based on the variables sex and comorbid diagnosis of cannabis use disorder. In the case of sex, the results show that significant differences were found in the case of DEA, OEA, PEA and POEA for men and in no NAE for women. In all of these compounds, plasma levels were higher in the SUD + SSRI antidepressant group (these differences were maintained after the Sidak correction test only in the case of DEA). In the case of the use of cannabis as a comorbid diagnosis, the results show that significant differences have been found in the case of DEA, LEA, OEA, PEA and POEA. Plasma levels of these NAEs were all higher in the SUD + SSRI antidepressant group (these differences were maintained after the Sidak correction test only in the case of PEA). Interestingly, no statistically significant difference was found in the non-cannabis-abusing group. These findings suggest that the influence of antidepressants on NAE levels in subjects who consult for alcohol or cocaine abuse may vary depending on sex and comorbid cannabis use.

### 2.4. Plasma Concentrations of Acylethanolamides Based on Type of Substance Use Disorder and Antidepressant Treatment

Since our SUD population were patients diagnosed with primary alcohol (AUD), cocaine (CUD), or both (AUD + CUD) use disorder, we further stratified them to analyze the effects of antidepressants on plasma concentrations of acylethanolamides. For this purpose, raw data for acylethanolamides were log10-transformed to ensure statistical assumptions of the two-way ANCOVA, with a type of SUD and antidepressant treatment as factors, while controlling for sex, age, and BMI. Figure 2 and Appendix A show the back transformation of the estimated marginal means and 95% CI of the PEA, POEA, SEA, OEA, LEA, DGLEA, AEA, and DEA based on a diagnosis of type SUD and antidepressant treatment. We defined three groups according to the diagnosis of lifetime SUD and SSRI-type antidepressant use: AUD (alcohol use disorder, with/without antidepressants) group; CUD (cocaine use disorder, with/without antidepressants) group; and AUD + CUD (A + C, with/without antidepressants) group. This analysis shows that only POEA concentrations were higher in the AUD group than in the CUD group, and only POEA and LEA concentrations were higher in the AUD + SSRI antidepressant group than in the CUD + SSRI antidepressant group. Regarding drug group, we observed drug-induced differences in POEA [F (3.211) = 3.818, *p* = 0.024], SEA [F (3.231) = 12.015, *p* < 0.001], DGLEA [F (3.332) = 7.136, *p* = 0.001], AEA [F (3.332) = 9.027, *p* < 0.001] and DHEA [F (3.332) = 8.369, *p* <0.001], derived of a decrease in the concentrations observed in the AUD group. After the Sidak correction test, statistical significance was maintained in the case of SEA, DGLEA, AEA and DHEA (*p* < 0.0057) (Figure 2). Considering the use of antidepressants, the only general effect observed was an increase in the concentrations of PEA. However, the significant effects of PEA [F (1.231) = 3.818, *p* = 0.014] for antidepressant treatment were not maintained after the Sidak correction test. Regarding drug group x antidepressant treatment interaction, we observed differences only in the case of AEA [F (2.332) = 3.486, *p* = 0.037]. Overall, again, the antidepressant treatment was not a major cause for differences in circulating acylethanolamides.

### 2.5. Plasma Concentration of Acylethanolamides as Predictors of Antidepressant Treatment

As we have described previously [29], logistic regression and ROC analysis of plasma NAEs demonstrated the utility of measuring these lipid mediators for the discrimination of patients with SUD from healthy control subjects (see Appendix A). Following this rationale, an additional logistic regression model for distinguishing patients with SUD without antidepressant treatment and patients with SUD with antidepressant treatment was performed using all acylehanolamides (log10-transformed concentrations), age, BMI, and sex (Appendix A). In this case, the ROC analysis indicated a poor discriminative power of the model (AUC = 0.669 (95% CI = 0.5766–0.7624), *p* < 0.001, Figure 3), which confirms that antidepressant treatment has no impact on circulating levels of NAEs in SUD patients.

## 3. Discussion

The main finding of the present study is the failure of antidepressant therapy to further increase the elevated levels of circulating NAEs that characterize patients with substance use disorder. This is a relevant finding in the context of the validation of NAEs as clinical biomarkers of SUD since defining interventions that might alter circulating NAEs is essential to establish their clinical utility. In addition, these results might give clues on some of the mechanisms underlying the loss of efficacy of antidepressants in dual depression patients [19]. SSRI-type antidepressant treatment in mild depression was described to be associated with increased circulating NAE concentrations [31] and the inability of SSRI treatment to exert this effect on substance use disorder patients might account for the reduced efficacy of SSRI in dual depression. However, our results also suggest that when assessing the influence of antidepressants on the concentrations of NAEs in this population, both the sex of the patient and the consumption of cannabis must be taken into account.

Addiction is a chronic relapsing disorder where multiple homeostatic systems are dysregulated, especially those related to the perception of pleasure and the subsequent emergent hedonism. Over the last years, the contribution to addiction of several signaling systems involved in hedonic homeostasis, including dopaminergic, opioid, glutamate, nociceptin, or NPY has been characterized [37,38]. Both, brain-released and plasma-circulating endocannabinoids/NAEs have been incorporated into this list, thanks to the study of chronic disorders such as alcohol use disorders or obesity, where the contribution of NAEs to the hedonic dysregulation has been undoubtedly established [16,17,39,40]. Of special interest is the fact that some of these NAEs, such as OEA, can restore anhedonic responses reflected as the failures of dopaminergic transmission to control responses for natural reinforcers such as caloric foods [16]. This reversion of anhedonia has been also confirmed when the origin of the anhedonic/depressive response is excessive alcohol consumption [32,33,39]. Thus, it seems clear that the circulating NAEs reflect a body’s attempt to restore multiple homeostasis failures, including mood and hedonism.

The fact that antidepressant treatment increases the circulating levels of NAEs in mild depression supports the notion that this enhanced NAE response engages with monoaminergic systems involved in hedonic/affective responses [16,31]. If we assume that enhanced circulating NAEs might be part of a general recovery response, the increased concentrations of these lipids observed in SUD patients might reflect the body’s attempt to restore the normality altered by the actions of abused drugs. In fact, this response attenuates with the length of abstinence, that is, the longer the time without consuming drugs, the more normal NAE profile is found in SUD patients [41]. Interestingly this response seems to be limited. In our cohort, nominally significant differences were found with higher levels of DEA and PEA in patients taking antidepressants compared to those not taking them. These differences were only marginally significant for OEA and POEA. In the ANCOVA analysis, we found a marginally significant group effect for DEA and OEA, and when the cohorts were divided into alcohol use disorder and cocaine use disorder, taking antidepressants also showed a nominally significant group effect for PEA. All these NAEs are well-described anti-inflammatory, anti-obesity, or reward-normalizing compounds. PEA was identified in the 1950s with anti-allergic and anti-inflammatory properties and has been reported to have a neuroprotectant role in several models of neuropsychiatric diseases [42]. OEA reduces alcohol self-administration and relapse as well as neuroinflammation induced by acute alcohol administration [6,32]. Less known is the role of DEA, although it has also been described that it can reduce LPS-induced mRNA expression of inflammatory mediators [43].

Taken together, these data could suggest that depressive patients with SUD would not respond to antidepressant treatment through a mechanism that involves a significant increase in the already elevated levels of NAEs. Moreover, this finding might be relevant if this blunted response of NAEs to antidepressant therapy has a role in the low clinical efficacy of SSRIs in dual depression. The experiments on the recovery of hedonic response by the injection of NAEs (basically OEA) in obese animals, demonstrating the normalization of the blunted dopamine response to natural reinforcers associated with long-term consumption of highly caloric diets [16], clearly support this hypothesis. However, this hypothesis needs to be confirmed in models of resistance to SSRI-induced antidepressant effects in depressed animals exposed to abused drugs.

Another interesting aspect of the present study is the descriptive nature of the population of users of substances of abuse that take SSRI antidepressants. Several findings are remarkable. First, we have observed a higher frequency of antidepressant prescriptions in women who use drugs than in men. This fact coincides with that reported in other studies on gender differences in the prescription of antidepressant medication in primary care, having been proposed as a possible explanation for women’s tendency to demand more medication, or the more open verbalization of their symptoms that facilitates physicians’ prescription of symptoms-relieving psychiatric medication [44]. Second, as expected, patients who take antidepressants have more psychiatric comorbidity and take more other classes of psychotropic drugs. This would also support the fact of the proven poly-pharmacy in the case of patients with psychiatric diagnoses [45]: when a psychotropic drug is started, there are many more possibilities to start progressively more and more. Third, the only drug of abuse whose use was significantly higher in the cohort taking antidepressants was cannabis. This finding could be in line with the old hypothesis of self-medication: some patients demanding antidepressants would consume the addictive substance in a potential use for “self-treat” depressive symptoms [46]. Further research is needed to unveil whether these gender differences are related to sex-dependent alterations in NAE biochemistry.

The present study has some limitations. First, we have measured only a cross-sectional assessment of the population with SUD, making it difficult to extrapolate conclusions of causality or measure the evolution of NAE levels over time. Secondly, apart from substance abuse and taking SSRI-type antidepressants, our population also has many other comorbidities and also takes other medications for this reason. Third, the severity of psychiatric symptoms was not consistently assessed, although subjects were stable enough to undergo the PRISM structured interview and sign the informed consent document. Furthermore, differences in gender or psychiatric comorbidity in both groups may raise doubts about the real reason for the differences found, which should be clarified in subsequent studies. It should also be noted that only slightly more than half of the patients with SUD who took antidepressants in our study were diagnosed in the psychiatric evaluation of a mood disorder, so the extrapolation of results to the population with dual depression must be done carefully. Finally, we do not have access to the last exact SSRI they were taking when the blood was collected. For all these reasons, we consider it necessary to continue investigating this interesting question, with larger samples and, if possible, with active follow-up. If possible, these new studies should try to elucidate the exact effects that alterations of NAEs have on the brains of addicted subjects. For example, other studies have described that in the healthy population, there is an inverse association between the circulating levels of several endocannabinoids and the availability of the cannabinoid receptor type 1 (CB1R) in certain brain areas and that this phenomenon is not replicated in subjects with other psychiatric disorders such as psychosis [47]. This and other questions to be solved will mark the future directions of research on the endocannabinoid system and substance use disorders.

## 4. Materials and Methods

### 4.1. Participants and Recruitment

In order to analyze the effect of antidepressant treatment on the circulating levels of acylethanolamides we recruited 333 abstinent substance use disorder Caucasian patients from both, the Centro Provincial de Drogodependencias (Málaga, Spain) and the outpatient alcohol program at Hospital Universitario 12 de Octubre (Madrid, Spain). Control participants (175 healthy volunteers) were included from databases of healthy subjects willing to participate in medical research projects from multidisciplinary staff working at the Hospital Regional Universitario de Malaga (Malaga, Spain), Hospital Universitario 12 de Octubre and Universidad Complutense de Madrid (Madrid, Spain). The total sample size was 506, according to the calculations performed with the GPower 3.1.9.2 program, considering an effect size of d = 0.3, patients/controls ratio of 2, and alpha = error of 0.05.

The inclusion criteria for participants to be eligible for the present study were the following: ≥18 years to 65 years of age and abstinence from alcohol (at least 4 weeks) and/or cocaine (at least 2 weeks) in the evaluation moment (screened at the clinical setting by self-report and confirmed by breath alcohol test or urine test for cocaine use). The exclusion criteria included: a personal history of long-term inflammatory diseases or cancer, cognitive or language limitations, pregnant or breast-feeding women, and infectious diseases. Prescription of antidepressant medication was obtained from clinical records and only patients using SSRI-type antidepressants were included. These medications could include fluoxetine, sertraline, citalopram, escitalopram, fluvoxamine, and paroxetine. SSRI antidepressants are the most prescribed in Spain and their influence on the levels of NAEs in depressed patients has been described in a previous work [31]. Regarding the control group, participants with psychiatric disorders or psychotropic drug consumption were also excluded.

### 4.2. Ethics Statements

Written informed consent was obtained from each participant after a complete description of the study. All the participants had the opportunity to discuss any questions or issues. The study and protocols for recruitment were approved by the Ethics Committee of the Hospital Regional Universitario de Málaga (PND2018I033, approved 25 October 2018) in accordance with the Ethical Principles for Medical Research Involving Human Subjects adopted in the Declaration of Helsinki by the World Medical Association (64th WMA General Assembly, Fortaleza, Brazil, October 2013) and Recommendation No. R (97) 5 of the Committee of Ministers to Member States on the Protection of Medical Data (1997), and the Spanish Data Protection Act (Regulation (EU) 2016/679) [48] of the European Parliament and the Council 27 April 2016 on the protection of natural persons concerning the processing of personal data and on the free movement of such data, and repealing Directive 95/46/EC (General Data Protection Regulation) [49]. All collected data were given code numbers in order to maintain privacy and confidentiality.

### 4.3. Clinical Assessments

Both psychiatric evaluation and blood extraction were performed in the morning times. SUD and other psychiatric disorders, including depression, were diagnosed according to the DSM-IV-TR criteria (APA, 2000) using the Spanish version of the Psychiatric Research Interview for Substance and Mental Disorders (PRISM). PRISM is a semi-structured interview with good psychometric properties in the evaluation of SUD and the main psychiatric co-morbid disorders related to the substance use population [18,50]. Healthy controls were examined with the Composite International Diagnostic Interview (CIDI) of the World Health Association.

### 4.4. Collection of Plasma Samples

Prior to the psychiatric interviews, blood samples were obtained in the morning after fasting. All participating subjects were summoned at 8:30 a.m. when the extraction was performed. Venous blood was extracted into 10 mL K2 EDTA tubes (BD, Franklin Lakes, NJ, USA) and immediately processed to obtain plasma for 8:30–12 h. Blood samples were centrifuged at 2200× *g* for 15 min (4 °C) and individually assayed to detect infectious diseases by four commercial rapid tests for HIV, hepatitis B, and hepatitis C (Strasbourg, Cedex, France) and SARS-CoV-2 (Bio-Connect, Huissen, the Netherlands). Finally, plasma samples were individually characterized, registered, and stored at −80 °C until further analyses.

### 4.5. Quantification of Acylethanolamides in Plasma

The analysis of acylethanolamides in plasma was performed by a HPLC-MS method previously described [51]. The following acylethanolamides were quantified: palmitoylethanolamide (PEA), stearoyl ethanolamide (SEA), oleoylethanolamide (OEA), palmitoleoylethanolamide (POEA), arachidonoyl-ethanolamide (AEA), linoleoylethanolamide (LEA), docosahexaenoylethanolamide (DHEA), di-homo-γ-linolenylethanolamide (DGLEA), and docosatetraenoyl-ethanolamide (DEA).

Briefly, aliquots of 0.5 mL of human plasma were transferred to 12 mL glass tubes, spiked with deuterated internal standards, diluted with 0.1 M ammonium acetate buffer (pH 4.0), and extracted with a tert-butyl methyl ether. The dry organic extracts were reconstituted in 100 μL of a mixture water:acetonitrile (10:90, *v*/*v*) with 0.1 percent formic acid (*v*/*v*) and transferred to HPLC vials. Twenty microliters were injected into the LC/MS-MS system. An Agilent 6410 triple quadrupole (Agilent Technologies, Wilmington, DE, USA) equipped with a 1200 series binary pump, a column oven, and a cooled auto-sampler (4 °C) was used. Chromatographic separation was carried out with an ACQUITY UPLC C18-CSH column (3.1 × 100 mm, 1.8-μm particle size) (Waters, Yvelines Cedex, France) maintained at 40 °C with a mobile phase flow rate of 0.4 mL/minute. The composition of the mobile phase was: A, 0.1 percent (*v*/*v*) formic acid in water; B, 0.1 percent (*v*/*v*) formic acid in acetonitrile. Quantification was performed by isotope dilution. Deuterated internal standards were obtained from Cayman Chemical (Ann Arbor, MI, USA), and solvents were from Merck (Darmstadt, Germany).

### 4.6. Statistical Analysis

Date in Table 1, Appendix A were expressed as the number and percentage of the subject (n (%)), mean and standard deviation (SD), or median and interquartile range (median (IQR)). Statistical differences in categorical variables were evaluated with the chi-square test or Fisher’s exact test, whereas differences in continuous variables were evaluated with the Student’s *t*-test for a normal distribution or the Mann–Whitney U test for a non-normal distribution.

Analysis of covariance (ANCOVA) (Figure 1 and Figure 2) was used to evaluate the main effects and interaction of primary independent variables (group/subgroup factor) (i.e., control and SUD; SUD and SUD + Antidepressant treatment) on NAE concentrations while adjusting for age, BMI, and sex as covariates. Raw data for NAE concentrations were log10-transformed because their distribution was positively skewed to ensure statistical assumptions of the ANCOVA. It was verified that after the logarithmic transformation, the new values followed a normal distribution that allowed the test to be carried out. Post hoc comparisons for multiple comparisons were performed using Sidak’s correction test. The estimated marginal means and 95% confidence interval (95% CI) of log10-transformed NAE concentrations were back-transformed in the figures. Receiver operating characteristics (ROC) analyses were performed to evaluate the discriminative power of binary logistic regression models through the area under the curve (AUC). We consider in line with the existing literature a lower limit of this equal to 0.75 to consider an adequate discriminative power of the model [52]. In addition, the resulting probability data from these models were compared between groups/subgroups using Student’s *t*-test or Mann–Whitney U test.

The GraphPad Prism version 5.04 (GraphPad Software, San Diego, CA, USA) and IBM SPSS Statistics version 22 (IBM, Armonk, NY, USA) were used for the statistical studies. A *p*-value of less than 0.05 was considered statistically significant. In the case of post hoc comparisons for multiple comparisons using Sidak’s correction test, a *p*-value of less than 0.0057 was considered statistically significant (1 − [1 − 0.05]^1/9^).

## 5. Conclusions

Our study shows that patients treated with SSRI-type antidepressants do not exhibit greater concentrations of circulating NAEs than substance abuse patients not treated with these medications. Antidepressant-treated patients retained the enhanced circulating concentrations of NAEs derived from drug consumption when compared to the control population. These findings support the notion of using enhanced NAE production as biomarkers of SUD. In addition, the possibility that a blunted NAE response to antidepressant therapy might be related to the loss of efficacy of SSRI in dual depression is an attractive hypothesis that needs to be addressed in future studies.

## Figures and Tables

**Figure 1 ijms-24-14788-f001:**
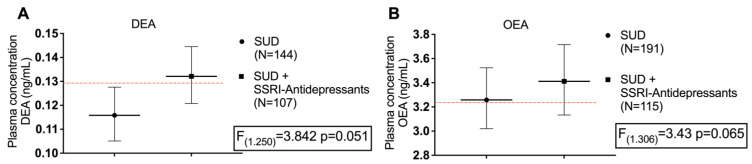
Plasma concentrations of (**A**) docosatetraenoyl-ethanolamide (DEA) and (**B**) oleoylethanolamide (OEA) in patients with substance use disorder (SUD) not using antidepressants, and patients of SUD using antidepressants. Red line represents the mean plasma concentration of the control healthy population. Data were analyzed by one-way analysis of covariance (ANCOVA). Dots are estimated as marginal means and 95% confidence intervals.

**Figure 2 ijms-24-14788-f002:**
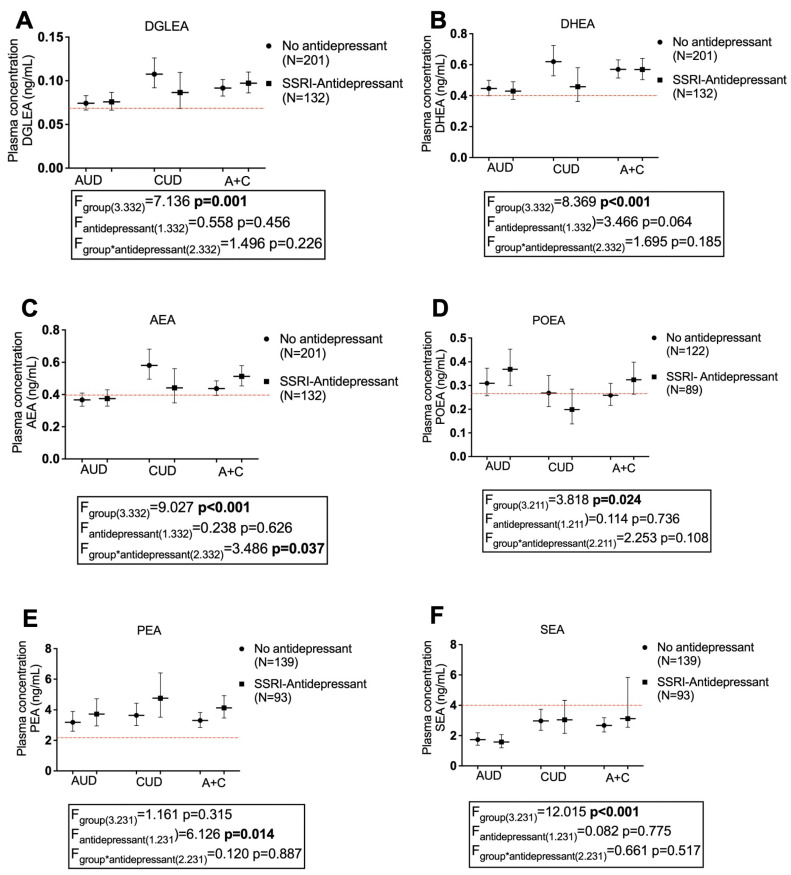
Plasma concentrations of (**A**) dihomo-γ-linolenylethanolamide (DGLEA), (**B**) docosahexaenoylethanolamide (DHEA), (**C**) arachidonoyl-ethanolamide (AEA), (**D**) palmitoleoylethanolamide (POEA), (**E**) palmitoylethanolamide (PEA), and (**F**) stearoyl-ethanolamide (SEA) in patients with substance use disorder (SUD) not using antidepressants, and patients with SUD using antidepressants classified on the basis of their diagnosis of alcohol use disorder (AUD), cocaine use disorder (CUD) or AUD + CUD. Red line represents the mean plasma concentration of the control healthy population. Data were analyzed by two-way analysis of covariance (ANCOVA). Dots are estimated as marginal means and 95% confidence intervals.

**Figure 3 ijms-24-14788-f003:**
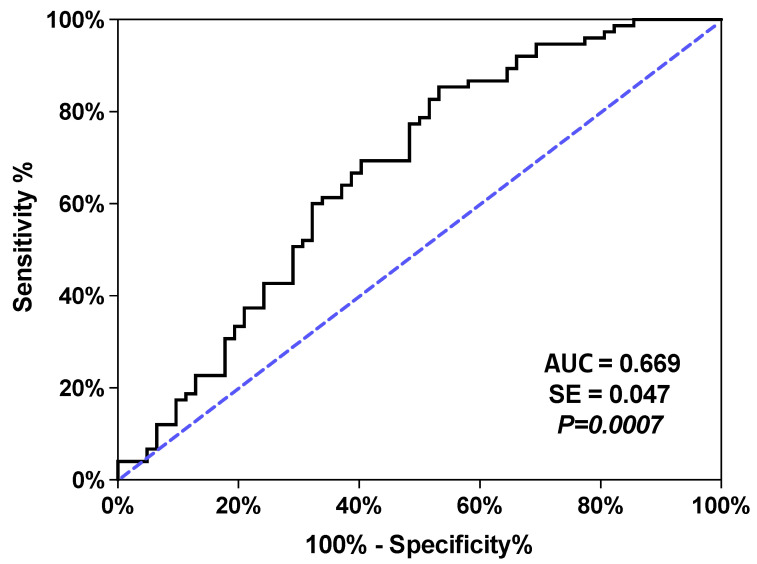
ROC analyses for plasma concentrations of acylethanolamides-based multivariate full models of predictive analysis of discrimination. SE: Standard Error.

**Table 1 ijms-24-14788-t001:** Sociodemographic and clinical characteristics of SUD (no antidepressant) versus SUD + SSRI antidepressant patients.

Variable		SUD Group (N = 201)	SUD + SSRI Antidepressant (N = 132)	*p* Value
SEX [N (%)]	**Men**	169 (84.1)	97 (73.5)	**0.018 ^a^**
	**Women**	32 (15.9)	35 (26.5)	
AGE (mean ± SD)		43.81 ± 11.22	43.03 ± 11.10	0.528 ^b^
BMI (mean ± SD)		26.15 ± 4.79	26.67 ± 4.96	0.338 ^b^
Psychiatric comorbidity [N (%)]	**Mood Disorders**	71 (35.3)	72 (54.5)	**0.001 ^a^**
	**Anxiety Disorders**	50 (24.9)	45 (34.1)	0.069 ^a^
	**Psychotic Disorders**	19 (9.5)	17 (12.9)	0.325 ^a^
	**Personality Disorders**	41 (20.4)	44 (33.3)	**0.008 ^a^**
	**ADHD**	32 (15.9)	34 (25.8)	**0.028 ^a^**
	**>2 psychiatric disorders**	122 (60.7)	110 (83.3)	**<0.001 ^a^**
Substance use disorders [N (%)]	**Alcohol**	165 (82.1)	116 (87.9)	0.155 ^a^
	**Cocaine**	122 (60.7)	79 (59.8)	0.877 ^a^
	**Cannabis**	39 (19.4)	39 (29.5)	**0.033 ^a^**
	**>2 substances**	102 (50.7)	76 (57.6)	0.222 ^a^
Psychotropic medication [N (%)]	**Antidepressants**	-	132 (100.0)	**-**
	**Anxiolytics**	83 (41.3)	73 (55.3)	**0.027 ^a^**
	**Antipsychotics**	18 (9.0)	22 (16.7)	**0.034 ^a^**
	**Disulfiram**	71 (35.3)	62 (47.0)	**0.034 ^a^**
SUD duration years [median (IQR)]	**AUD**	10 (4–16.25)	10 (4–18.5)	0.894 ^c^
	**CUD**	6 (2–12)	5 (2–11)	0.640 ^c^
Days of abstinence [median (IQR)]	**AUD**	70 (0–240)	64.5 (7.75–157.5)	0.991 ^c^
	**CUD**	30 (0.75–157.5)	15 (0.25–90)	0.208 ^c^

(^a^) *p*-value from chi-square test; (^b^) *p*-value from Student’s *t*-test. (^c^) *p*-value from Mann–Whitney U test, *p*-value in bold indicates a statistically significant difference. Abbreviations: BMI = body mass index; IQR = interquartile range; SD = standard deviation; SUD = substance use disorder; AUD = alcohol use disorder; CUD: cocaine use disorder.

**Table 2 ijms-24-14788-t002:** Plasma concentrations of acylethanolamides according to SUD or SUD + SSRI antidepressant group.

NAEs ^a^	SUD (N = 201)	SUD + SSRI Antidepressant (N = 132)	U-Statistic	*p* Value ^b^
**AEA** median (IQR)	0.44 (0.30–0.64)	0.44 (0.31–0.65)	12,834.00	0.615
**DEA** median (IQR)	0.12 (0.09–0.16)	0.14 (0.10–0.19)	**6189.00**	**0.008**
**DGLEA** median (IQR)	0.08 (0.06–0.13)	0.08 (0.06–0.12)	12,984.00	0.742
**DHEA** median (IQR)	0,54 (0.36–0.74)	0.51 (0.38–0.70)	12,441.00	0.337
**LEA** median (IQR)	1.07 (0.86–1.35)	1.12 (0.86–1.42)	12,117.00	0.181
**OEA** median (IQR)	3.26 (2.47–4.26)	3.66 (2.59–4.76)	9518.00	0.051
**PEA** median (IQR)	3.02 (2.22–4.96)	4.12 (2.38–6.60)	**5209.00**	**0.012**
**POEA** median (IQR)	0.27 (0.18–0.42)	0.33 (0.23–0.50)	4585.00	0.054
**SEA** median (IQR)	2.07 (1.21–4.22)	1.97 (1.47–4.15)	6298.00	0.740

(^a^) Plasma concentrations of NAEs expressed in ng/mL (^b^) *p*-value from Mann–Whitney U test. *p*-value in bold indicates a statistically significant difference. Abs.: NAEs: *N*-acylethanolamides; SUD: substance use disorders; IQR: interquartile range; AEA: arachidonoyl-ethanolamide; DEA: docosatetraenoyl-ethanolamide; DGLEA: di-homo-γ-linolenylethanolamide; DHEA: docosahexaenoylethanolamide; LEA: linoleoylethanolamide; OEA: oleoylethanolamide; PEA: palmitoylethanolamide; POEA: palmitoleoylethanolamide; SEA: stearoyl-ethanolamide.

**Table 3 ijms-24-14788-t003:** Plasma concentrations of acylethanolamides according to SUD or SUD + SSRI antidepressant group and sex.

NAEs ^a^	Men			Women		
	SUD	SUD + SSRI-AD	*p* Value ^b^	SUD	SUD + SSRI-AD	*p* Value ^b^
**AEA** median (IQR)	0.45 (0.31–0.64)	0.52 (0.36–0.73)	0.066	0.40 (0.26–0.55)	0.34 (0.26–0.43)	0.294
**DEA** median (IQR)	0.12 (0.10–0.17)	0.16 (0.12–0.22)	**0.001**	0.11 (0.08–0.15)	0.11 (0.09–0.15)	0.647
**DGLEA** median (IQR)	0.08 (0.06–0.13)	0.09 (0.06–0.12)	0.391	0.08 (0.06–0.13)	0.08 (0.06–0.10)	0.641
**DHEA** median (IQR)	0.54 (0.36–0.74)	0.53 (0.41–0.72)	0.980	0.52 (0.39–0.67)	0.42 (0.30–0.57)	0.077
**LEA** median (IQR)	1.07 (0.86–1.35)	1.09 (0.89–1.46)	0.101	1.09 (0.87–1.24)	1.20 (0.73–1.37)	0.851
**OEA** median (IQR)	3.24 (2.47–4.29)	3.86 (2.62–5.15)	**0.017**	3.42 (2.49–4.00)	3.17 (2.55–4.30)	0.829
**PEA** median (IQR)	3.08 (2.23–4.98)	4.31 (2.28–6.92)	**0.012**	2.83 (1.87–4.89)	3.79 (2.54–4.49)	0.583
**POEA** median (IQR)	0.25 (0.18–0.38)	0.34 (0.24–0.43)	**0.035**	0.38 (0.18–0.54)	0.32 (0.22–0.56)	0.859
**SEA** median (IQR)	2.06 (1.21–4.36)	2.26 (1.54–4.55)	0.323	2.62 (1.19–3.67)	1.49 (1.19–1.90)	0.284

(^a^) Plasma concentrations of NAEs expressed in ng/mL (^b^) *p*-value from Mann–Whitney U test. *p*-value in bold indicates a statistically significant difference. Abs.: NAEs: *N*-acylethanolamides; IQR: interquartile range; AEA: arachidonoyl-ethanolamide; DEA: docosatetraenoyl-ethanolamide; DGLEA: di-homo-γ-linolenylethanolamide; DHEA: docosahexaenoylethanolamide; LEA: linoleoylethanolamide; OEA: oleoylethanolamide; PEA: palmitoylethanolamide; POEA: palmitoleoylethanolamide; SEA: stearoyl-ethanolamide.

**Table 4 ijms-24-14788-t004:** Plasma concentrations of acylethanolamides according to SUD or SUD + SSRI antidepressant group and comorbid use of cannabis.

NAEs ^a^	Cannabis +			Cannabis −		
	SUD	SUD + SSRI AD	*p* Value	SUD	SUD + SSRI AD	*p* Value ^b^
**AEA** median (IQR)	0.46 (0.27–0.64)	0.56 (0.33–0.73)	0.215	0.44 (0.30–0.63)	0.42 (0.30–0.59)	0.732
**DEA** median (IQR)	0.10 (0.08–0.15)	0.16 (0.12–0.23)	**0.022**	0.12 (0.10–0.16)	0.13 (0.10–0.18)	0.108
**DGLEA** median (IQR)	0.09 (0.05–0.15)	0.10 (0.07–0.13)	0.641	0.08 (0.06–0.12)	0.08 (0.06–0.11)	0.783
**DHEA** median (IQR)	0.54 (0.36–0.77)	0.54 (0.41–0.72)	0.865	0.54 (0.37–0.74)	0.49 (0.38–0.65)	0.250
**LEA** median (IQR)	0.99 (0.88–1.25)	1.35 (0.92–1.64)	**0.032**	1.10 (0.85–1.38)	1.07 (0.85–1.37)	0.907
**OEA** median (IQR)	2.93 (2.44–3.53)	3.86 (2.78–4.82)	**0.010**	3.38 (2.48–4.44)	3.62 (2.55–4.73)	0.354
**PEA** median (IQR)	2.57 (2.10–3.61)	4.71 (2.48–7.02)	**0.005**	3.30 (2.23–5.46)	4.01 (2.30–6.05)	0.230
**POEA** median (IQR)	0.18 (0.14–0.35)	0.35 (0.25–0.43)	**0.006**	0.28 (0.18–0.43)	0.33 (0.22–0.50)	0.320
**SEA** median (IQR)	3.21 (1.21–4.65)	2.70 (1.78–6.57)	0.432	2.00 (1.21–4.01)	1.73 (1.38–3.34)	0.649

(^a^) Plasma concentrations of NAEs expressed in ng/mL (^b^) *p*-value from Mann–Whitney U test. *p*-value in bold indicates a statistically significant difference. Abs.: NAEs: *N*-acylethanolamides; IQR: interquartile range; AEA: arachidonoyl-ethanolamide; DEA: docosatetraenoyl-ethanolamide; DGLEA: di-homo-γ-linolenylethanolamide; DHEA: docosahexaenoylethanolamide; LEA: linoleoylethanolamide; OEA: oleoylethanolamide; PEA: palmitoylethanolamide; POEA: palmitoleoylethanolamide; SEA: stearoyl-ethanolamide.

## Data Availability

Not applicable.

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
