# Peer review of "Antidepressant Medication Does Not Contribute to the Elevated Circulating Concentrations of Acylethanolamides Found in Substance Use Disorder Patients"

_ijms, 2023, doi:10.3390/ijms241914788_

Round 1

Reviewer 1 Report

Dear authors,

The following manuscript describes the absence of correlation between elevated circulating concentrations of acylethanolamides in substance use disorders patients and antidepressant medication. I consider some major changes before considering the publication of this paper in IJMS.

At first, the text and also the English of this manuscript needs to be carefully checked. Multiple English spelling mistakes are observed in the text. For example:

Line 54: intae instead of intake.

Line 58: After the references 5 and 6 a dot is missed.

Lines 112-117: The absence of any dot in the paragraph makes some difficult to follow the aim of the study. Please, rewrite to clarify it.

Line 115: Om instead on?

Line 141: That instead of than.

Line 145: Sentence is incomplete when referring to initial analysis and concentrations.

Line 195: Extra dot after the DHEA information.

Line 200: Onl instead of only.

Line 231: Se instead of see and which Supplementary Figure is referred?

Line 285: y instead of and.

The authors suggest the use of some abbreviations but following the text, these abbreviations are not always used. For instance:

Line 53 authors suggest the use of SUD instead of Substance Use Disorders but this abbreviation is not used in line 73, 84, 92, 95, 108, etc. Please, check,

In line 65, I would like to ask to the authors what do they mean when they use the expression “true endocannabinoids”? Is a common expression in the field? If so, please cite some reference to defend it.

In Section 2. Results, there is a subdivision of 3.2, 3.3 and 3.4 instead of 2.2, 2.3 and 2.4.

In Table 1, authors describe that the study presents more men than women. Could you demonstrate that results are the same in both genders?

In Section 2.2 (referred as 3.2), authors describe some changes in DHEA and SEA but regarding to Table 2, these changes are observed in DEA and PEA.

Why authors use mainly use ANCOVA (parametric test) but in Table 2 and Supplementary Table 3 use a non-parametric test?

I consider that Figures will be more representative if authors change bar plots for dot plots. Bar plots do not help to understand the variability of the population.

In Table 2, how authors can explain that the lower IQR point is greated that the higher IQR point for SUD+Antidepressant in OEA? In addition, I suggest authors to add the units of the results in the table.

I suggest authors to better explain what do they mean in lines 234-237 when they suggest that the ROC analysis suggest a poor discriminative power but the p value is lower than 0.05. Please, clarify this part.

Authors are used samples collected in the range of 8-12h. As it is described (and observed in Figure 1) in https://www.ncbi.nlm.nih.gov/pmc/articles/PMC7001881/, AEA changes in this range of 8-12h. So, these results may be taken into account at the time to compare samples collected at different time hours. I suggest authors to demonstrate that selecting a subgroup of samples but all of them collected at the same time the results are the same. If not, it may seem that the absence of changes is due to the collection time variability.

I suggest authors to show a chromatogram of all the different analytes at low concentrations.

The References section must be carefully checked. Some mistakes (e.g., absence of doi, some journals abbreviated and some no, some doi underlined and some no…) are observed. And also, reference 41 is started with the Journal name. Please, carefully check it.

Dear authors,

As it is described in the Comments and Suggestions for Authors Section, English must be carefully checked and improved.

Reviewer 2 Report

Substantive remarks:

The group of people covered by the study, apart from substance abuse and taking selective serotonin reuptake inhibitors, also has many other comorbidities. These people also take other medications for this reason (Table 1).

To draw valid conclusions, studies should not be conducted on such a diverse group of patients.

The presentation of the results of biochemical tests (plasma concentration of acylethanolamides) is not consistent with the clinical characteristics of the study group. Why are the concentrations of these compounds not shown in cannabinoid abusers?

The work had to include a clear division into groups in terms of substance use disorders (SUD).

What drugs from the SSRI group were the subjects taking?

When giving the characteristics of the study group, the means with SD or SEM should be given. This applies to e.g. age of the study group, BMI, etc.

Other notes:

There are many editorial errors in the text of the manuscript. For this reason, it requires careful proofreading for punctuation, capitalization, and spaces.

Among others:

• the text of the manuscript should be properly formatted (justified)

• line 4 - redundant dot after the title

• line 31 - unnecessary comma after [NAEs]

• there should be no space before the comma, eg line 116, 123

• line 119 - "Plasma Concentration" should be written in lower case

• unnecessary spaces: e.g. line 60, 134, 158, 187, 302, 327

• line 156 - the dot should be after "(Figure 1)

• line 161 - "(a)" should be superscripted

• "Table 1" (line 158) and "Table 2" (line 163) should be followed by dots instead of colons

• line 163 - no space before the table title

• line 174, 373 - unnecessary spaces before "ethanolamide"

• line 236, 387, 388 - remove the double brackets

• line 336 - should be "inflammatory" instead of "in-flammatory"

Figure 1 - poor graphic quality; Y-axis titles too small

Figure 2 - poor graphic quality; too small y-axis titles, too small font of text in frames

In addition, the record of the reference needs to be corrected as recommended in the Instructions for Authors (https://www.mdpi.com/journal/ijms/instructions).

References should be described as follows, depending on the type of work:

Journal Articles:

1. Author 1, A.B.; Author 2, C.D. Title of the article. Abbreviated Journal Name YearVolume, page range.

Books and Book Chapters:

·       Author 1, A.; Author 2, B. Book Title, 3rd ed.; Publisher: Publisher Location, Country, Year; pp. 154–196.

 ·       Author 1, A.; Author 2, B. Title of the chapter. In Book Title, 2nd ed.; Editor 1, A., Editor 2, B., Eds.; Publisher: Publisher Location, Country, Year; Volume 3, pp. 154–196.

An example of record (journal article):

Bowman, C.M.; Landee, F.A.; Reslock, M.A. Chemically Oriented Storage and Retrieval System. 1. Storage and Verification of Structural Information. J. Chem. Doc. 19677, 43-47; DOI:10.1021/c160024a013.

Reviewer 3 Report

This is a cross-sectional study, which has compared the levels of different bioactive lipids between SUD patients and healthy controls, and found elevated levels in plasma concentrations of the majority of acylethanolamides. However, those values did not differ between patients groups treated with or without SSRIs. This study is important because it points out acylethanolamides as potential markers of different substance abuse

Introduction: please, explain what are acylethanolamides and N-acylethanolamines. Why did you choose only SSRIs?

Materials and methods: for the inclusion criteria, please add that this refers to SUD patients, please provide information for how long were the patients required to be abstinent

4.3. Clinical Assessments 353 Clinical assessment was done following previously reported protocols. Please, describe shortly which assessments were conducted. Readers may not be familiar with previous work.

Results: Table 2. Please, provide full terms for all abbreviations

Please, add information on the duration of abstinence, i.e., whether the samples were taken during withdrawal, or later. Moreover, what was the severity of symptoms of psychiatric disorders, have the patients been in remission?

If patients were taking only SSRIs (and not other antidepressants), please replace antidepressants (from tables, and in other parts of the text) with SSRIs

Discussion: Authors are using the term: dual depression, but, according to table 1, only 55% of SUD participants who received SSRIs had mood disorders. So please, delete the term „dual depression”.

„Taken together, these data would suggest that depressive patients with SUD do not respond to antidepressant treatment by significantly rise the already elevated circulating levels of NAEs”-this sentence is unclear, please, correct

Please, add some data whether peripheral levels of acylethanolamides reflect those in brain

Round 2

Reviewer 2 Report

The authors responded to all my comments in the revised version of the manuscript ijms-2588255. Supplementing the text and making corrections had a positive impact on the quality of the article, both in terms of content and editorial.

However, some records of references still require minor corrections (e.g. items 8, 11, 17, 29, 47 - the title of the journal should be replaced with an abbreviation). I believe that this can be verified at the stage of editorial corrections.

My decision: accept in present form